# Fluid and Salt Balance and the Role of Nutrition in Heart Failure

**DOI:** 10.3390/nu14071386

**Published:** 2022-03-26

**Authors:** Christina Chrysohoou, Emmanouil Mantzouranis, Yannis Dimitroglou, Andreas Mavroudis, Kostas Tsioufis

**Affiliations:** First Cardiology Clinic, School of Medicine, University of Athens, 11527 Athens, Greece; mantzoup@gmail.com (E.M.); dimiyann@hotmail.com (Y.D.); andreasmavroudis89@gmail.com (A.M.); ktsioufis@gmail.com (K.T.)

**Keywords:** heart failure, nutrition, fluid balance, salt

## Abstract

The main challenges in heart failure (HF) treatment are to manage patients with refractory acute decompensated HF and to stabilize the clinical status of a patient with chronic heart failure. Beyond the use of medications targeted in the inhibition of the neurohormonal system, the balance of salt and fluid plays an important role in the maintenance of clinical compensation in respect of renal function. In the case of heart failure, a debate of opinion exists on salt restriction. Restricted dietary sodium might lead to worse outcomes in heart failure patients due to the activation of the neurohormonal system and malnutrition. On the contrary, positive sodium balance is the primary driver of water retention and, ultimately, volume overload in acute HF. Some recent studies reported associations of decreased salt consumption with higher readmission rates and increased mortality. Thus, the usefulness of salt restriction in heart failure management remains debated. The use of individualized nutritional support, compared with standard hospital food, was effective in reducing these risks, particularly in the group of patients at high nutritional risk.

## 1. Introduction

Heart failure (HF) remains one of the leading causes of recurrent hospitalizations, and it poses a large burden on healthcare systems. Even though significant advances in the field of pharmacologic and device therapy have been achieved, it still remains the single most common admission diagnosis in patients aged over 65 years of age [1]. The main challenges in HF treatment are to manage patients with refractory acute decompensated HF and to stabilize the clinical status of a patient with chronic heart failure. Beyond the use of medications targeted to the inhibition of the neurohormonal system, the balance of salt and fluid plays an important role in the maintenance of clinical compensation in respect of renal function [2]. Since a state of fluid overload is described as ‘‘congestive heart failure’’, heart function has received more attention than renal function, although the interaction between the heart and the kidneys plays a critical role in the progression of HF and the clinical outcome [3].

## 2. Fluid and Salt Balance in Chronic Heart Failure and Salt Consumption Recommendations

Current guidelines recommend restricting sodium intake, ranging from 1500–3000 mg per day, although a debate of opinion still exists regarding the maximum sodium intake for HF patients [4]. High sodium intake has been related with volume retention, high blood pressure, and increased cardiovascular morbidity. Data from studies introducing low sodium intake diets, such as the Dietary Approaches to Stop Hypertension (DASH) diet, present a significant decrease in all-cause mortality and the risk of adverse cardiac events [5,6]. There are some concerns in the case of heart failure patients, as they are usually advised to follow a low sodium diet; many of the people who consume a low sodium diet are more likely to have heart failure compared to the general population. Additionally, there is a concern that sodium restriction in a diet may be accompanied by a poor caloric intake and decreased nutritional status, which may confound heart failure outcomes. Restricted dietary sodium intake can lead to worse outcomes in heart failure patients for two main reasons. The first is that low sodium intake can activate the renin angiotensin aldosterone hormonal system (RAAS), leading to exacerbation of the symptoms of heart failure, and a decline in the glomerular filtration rate, even in the presence of a blockage of the hormonic system, which improves renal blood flow [7,8]. This is due to the negative effect of a low sodium intake on cardiac output and stroke volume, as well as the activation of the sympathetic nervous system (SNS) by the macula densa and the subsequent increase in vascular resistance. These mechanisms are more evident in profound clinical stages of HF. Another reason is that low sodium intake has undesirable effects on insulin resistance and serum lipids, factors that predispose patients to cardiovascular diseases [9,10,11]. A meta-analysis of 23 cohort studies and 2 clinical trials demonstrated that dietary sodium restriction did not improve the quality of life over the long term, did not reduce the readmission rate within the short term (≤30 days), and increased the readmission rate over the long term. Additionally, patients consuming diets with a moderate (2300 milligrams per day) sodium intake showed increased natriuresis and diuresis compared to patients with a low sodium intake [12]. However, reverse causation in those observational studies might have been a confounding factor.

## 3. Decompensated Heart Failure Fluid and Salt Balance

Congestion has a central role in patients with decompensated heart failure (HF) as it is related to increased morbidity and higher readmission rates, and leads to the vicious cycle of diuretic resistance. The primary mechanism of diuretic resistance in the context of decompensated HF and congestion is increased renal vein pressure, which decreases the arteriovenous pressure gradient, with a parallel increase in the interstitial pressure within the kidney, thereby decreasing renal blood flow, ultrafiltration pressure, and the glomerular filtration rate (GFR). Congestion also exacerbates diuretic resistance by decreasing intestinal absorption and stimulating the SNS and RAAS. Moreover, decreased liver function in patients with right heart failure and congestion leads to hypoalbuminemia, which can increase resistance to furosemide, the most commonly used diuretic. Thus, in a clinical course of patients with decompensated HF, the development of diuretic resistance and progressive cardiorenal dysfunction, despite escalating doses of loop diuretics, resulted in negative outcomes [13]. This is more evident in patients with predominant right HF with the clinical signs of peripheral edema, ascites, and fluid retention. In this setting, the right ventricle shows reduced inotropic reserve, causing a venous volume overload. Aggressive diuresis in these cases can further disrupt the sodium–fluid balance that the kidney attempts to maintain [14]. Furthermore, inotrope infusion to increase the output of the right ventricle, combined with diuretic therapy, seems in many cases insufficient, as those patients may also need a fluid challenge.

An important physiological principle of edema formation in acute HF is that a positive sodium balance serves as the primary driver of water retention and, ultimately, volume overload. Although studies have shown that sodium intake leads to the worsening of congestive symptoms, contradictory data have revealed that sodium restriction may not confer a clear benefit with respect to congestive symptoms and clinical outcomes, and it can lead to heightened neurohormonal activity [7,8]. As an extension of that concept, several studies in patients with acute HF with diuretic resistance have shown that the administration of hypertonic saline, given concomitantly with high doses of loop diuretics, may improve diuresis, renal function, and clinical outcomes in some cases, even with inotropic support [15,16]. It was hypothesized that a bolus infusion of hypertonic saline can mobilize fluid from the interstitial space to the intravascular compartment through osmotic forces, thereby restoring effective intravascular volume, enhancing renal blood flow, and improving the delivery of diuretic agents to the loop of Henle [17,18]. This hypothesis was also investigated in another study in which a slow continuous infusion of hypertonic saline (1.7% NaCl at a rate of 0.35 mL/min) resulted in clinical benefits similar to those of a bolus infusion. Data from Real World Use of Hypertonic Saline in Refractory Acute Decompensated Heart Failure show that the administration of hypertonic saline is associated with increased diuretic efficiency, fluid and weight loss, and a parallel improvement in metabolic factors with no additional adverse effects [19,20].

In the case of pleural effusion, the underlying physiology remains controversial. The fundamental model of pleural exchange in normal conditions involves filtration through the capillaries in the parietal pleura lining the chest wall and drainage of the pleural liquid via lymphatic stomata in the parietal pleura. However, the classical Starling equations do not take into account the electrolyte balance in the formation of the pleural fluid. According to the “chloride theory”, chloride is the key electrolyte for regulating the distribution of body fluid or water in each body compartment [21,22,23]. Under normal physiological conditions, chloride anion concentrations in the interstitial space are high when compared with serum [20,24]. A gradient in chloride concentrations between the interstitial and pleural spaces is responsible for the differential amounts of fluid in each compartment. In worsening HF, there is a high pleural chloride concentration compared with the lower chloride concentration in the normal physiological state, suggesting that chlorine has an active role in the formation of pleural fluid.

These findings suggest that, with appropriate patient selection, administration of hypertonic saline to an acute HF patient does not necessarily result in fluid retention or worsening pulmonary edema and hypoxemia. Additionally, an excessive correction of serum sodium should be avoided, and the recommended correction of >6 to 8 mmol/L per 24 h should be maintained in the hyponatremic patient [25,26]. Recent ESC guidelines for heart failure refer to cautious fluid administration in patients with AHF, especially in the setting of isolated right heart failure, to improve peripheral hypoperfusion with a class IIb indication [6].

Another issue is how to handle fluid overload in HF patients, where the management window between dehydration and overhydration is narrow. The main aim in this situation is to restore fluid balance without risking any further decrease in cardiac performance and stroke volume. In the cases of right heart failure, this balance is difficult to achieve, as low cardiac output in combination with venous congestion and decreased preload leads to progressive renal retention of salt and water. This causes expansion of extracellular space and progressive distention of the myocardium, with adverse effects on cardiac performance. In such cases, the restoration of fluid balance must be cautious to prevent any hemodynamic instability, or it may restore the osmotic gradient within the loop of Henle.

In the challenging scenario of acute decompensated HF refractory to all first line treatments with intravenous diuretics and combination of diuretics, ultrafiltration comes to the foreground as a viable solution for fluid removal. It represents a method of energetic withdrawal of isotonic fluid from the venous system, performed with or without the assistance of inotropes and vasospastic drugs. Despite the efficacy of the method in terms of forced fluid removal, when the ultrafiltration rate significantly exceeds the plasma refill rate, there is intravascular volume depletion, which in preload-dependent HF patients can lead to a further decrease in cardiac output. In such cases, continuous ultrafiltration has emerged as a decongestion method for patients with refractory decompensated heart failure. This method enables the energetic withdrawal of isotonic fluid from the venous system under a controlled rate according to the patient’s vital signs. Furthermore, continuous ultrafiltration may compromise excessive neurohormonal activation [27].

According to the available data and experiences, continuous ultrafiltration could be considered in patients with congestion refractory to intravenous diuretics and diuretic combinations, low urine output (<100 mL), impaired renal function, compromised function of the right ventricle, and frequent hospitalizations for decompensated HF. Termination of the procedure is indicated after the clinical resolution of congestion and after either persistent elevation of creatinine by more than 1 mg/dL compared to baseline or persistent hemodynamic instability [28,29,30,31,32,33,34].

## 4. Pathophysiological Mechanisms of Sodium Intake in Heart Failure

Nowadays, the theory that salt is universally deleterious, whereby excessive sodium may exacerbate acute heart failure and congestion, has been re-evaluated, as a restriction in salt intake seems to lead to a heightened sodium avidity signal. This results in neurohormonal activation and questionable differences in congestive symptoms. The role of chloride anion on salt-sensitive renal responses of tubuloglomerular feedback and renin release has been re-appraised [35], especially in the field of HF. A family of serine/threonine kinases (with-no-lysine [K] (WNK)) appear to act as critical regulators of electrolyte homeostasis in the actions of the RAAS and in the mechanisms where loop and thiazide diuretic agents work [36,37]. These observations led to the hypothesis that chloride may play a more important role in HF, even surpassing the role of sodium as a determinant of diuretic response, neurohormonal activation, and eventually the prognosis of those patients.

## 5. Precise Medicine in Fluid and Nutritional Aspects

HF is a systemic syndrome characterized by reduced cardiac output and/or high intracardiac pressure, compromising the heart’s ability to maintain adequate oxygenation of tissues. It is known that a failing heart relies on exogenous substrates for energy provision, resulting in ischemic injury and electrical vulnerability; thus, nowadays, the importance of dietary modification, beyond the preservation of uncontrolled weight loss, has attracted scientific and clinical interest. In the course of HF, a clinician has to face the challenge of cachexia or the obesity paradox.

Cachexia is a complex metabolic wasting syndrome, characterized by unintentional weight loss of edema-free mass, anorexia, inflammation, anemia, and low serum albumin. All these are predictors of adverse HF outcomes. Potential treatments include appetite stimulants (ghrelin antagonists, mirtazapine, and dronabinol); exercise; and anabolic agents, used in combination with nutritional supplements and anti-catabolic interventions, although the benefits and safety of these have not been proven [38].

Once HF is diagnosed, increased body weight has been associated with lower mortality and better prognosis, while significant weight loss appears to correlate with higher mortality rates. In the case of body weight loss, the focus should be on the quality of the weight lost, and whether it is adiposity or involves muscle mass. Hence, nutritional status may explain this paradox when non-sarcopenic obesity exists or in excessive lean mass loss. The co-existence of a metabolically healthy status may multiply the protective effect of increased weight.

Obesity has a prevalence of 30–40% in HF with a reduced ejection fraction (HFrEF) and 40–55% in HF with preserved ejection fraction (HFpEF), making this condition a challenge. According to the European Society of Cardiology (ESC), weight loss cannot be recommended in patients with moderate obesity, which is in line with the recommendation of the American Heart Association (AHA). In advanced obesity, weight loss manages symptoms and exercise capacity. To address this issue, in clinical practice, it is common to recommend a negative energy balance of 500–750 kcal/d or an absolute intake of 1200–1500 kcal/d for women and 1500–1800 kcal/d for men, aiming for a loss of at least 5–10% of the weight at baseline [39].

## 6. Nutritional Issues in Chronic HF

Micronutrient deficiencies are attributed to appetite loss, diuretic use, renal dysfunction, comorbidities that induce dietary restrictions or malnutrition (e.g., diabetes and depression), and medications that alter the gut microbiome. Epidemiological studies have detected low intakes of vitamin A, calcium, magnesium, selenium, and iodine in 20–30% of HF patients, and low vitamin D intake in 75% of them. Despite the growing evidence regarding micronutrient deficiencies in chronic HF, the lack of recognized nutritional strategies remains an unmet challenge in HF research. Although omega-3 fatty acids may reduce arrhythmias, stabilize contractile heart cells electrically, and increase heart rate variability, there is no new randomized clinical trial to justify their use. Thus, there is weak evidence for the clinical use of omega-3 supplementation in heart failure patients [40], considering the level of evidence of IIB in HFrEF and lack of recommendation in HFpEF [41]. In addition, although vitamin D deficiency has been shown to increase patients’ mortality by 2.8 times and deteriorates kidney function, there is no clear indication for supplementation beyond the 4000 IU/d in the case of renal dysfunction [42].

### Dietary Patterns

ESC guidelines focus on fluid intake, weight monitoring, and excessive salt and alcohol avoidance, accompanied by the general principles of CVD prevention (i.e., reduced consumption of saturated fatty acids; increased intake of fruits, vegetables, unsalted nuts and dietary fibers). A Mediterranean diet, fundamentally similar to the DASH diet, demonstrates positive correlations with HF progression in epidemiological studies, but additional RCTs/cohorts are needed [43,44]. In experimental models, olive oil, a primary substance in the Mediterranean diet, reduced infarct size and protected reperfused myocardium from oxidative damage in vivo. Hyperproteinic, low-carbohydrate, and low-fat diets also demonstrate positive effects on functional capacity; however, RCTs in secondary prevention are scarce [45,46,47]. Nutritional education seems to be important for HF patients in order to sustain fluid and salt balance and prevent cachexia through a balanced diet.

## 7. Conclusions

Dietary recommendations for heart failure management have traditionally focused on sodium and fluid restrictions. However, emerging data support the role of moderate sodium and fluid restriction, using a personalized approach. The use of individualized nutritional support, compared with standard recommendations, has shown effectiveness, especially in patients at high nutritional risk.

## Data Availability

Not applicable.

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
