# Peer review of "Fluid and Salt Balance and the Role of Nutrition in Heart Failure"

_nutrients, 2022, doi:10.3390/nu14071386_

Round 1
Reviewer 1 Report
The report is a brief review of salt homeostasis and nutrition in heart failure patients. It requires extensive editing as the concepts are obscured and the message confused by the language errors.
Lines 42-46 and 56-61 are difficult to understand due to grammatical errors.
The section regarding RHF (64-77) is important and could be expanded.
Section 77-95 discusses hypertonic saline and why it may be effective treatment to promote diuresis. In addition to the noted potential mechanisms, it may help restore the osmotic gradient within the Loop of Henle. This section could be expanded as the use of hypertonic saline in HF patients has not been widely adopted.
The “chloride theory” of fluid retention could be expanded (96-110, 133-144). I am unfamiliar with word “Chlorion” and that should be explained.
Ultrafiltration (118-131) remains a niche treatment. Are there advantages and data supporting its use?
The paragraph 159-167 should be re-written.
The report is well referenced.
Author Response
Reviewer 1
The report is a brief review of salt homeostasis and nutrition in heart failure patients. It requires extensive editing as the concepts are obscured and the message confused by the language errors.
We would like to Thank the Reviewer for the useful remarks
Lines 42-46 and 56-61 are difficult to understand due to grammatical errors.
Reply We have modified this section.
The section regarding RHF (64-77) is important and could be expanded.
Section 77-95 discusses hypertonic saline and why it may be effective treatment to promote diuresis. In addition to the noted potential mechanisms, it may help restore the osmotic gradient within the Loop of Henle. This section could be expanded as the use of hypertonic saline in HF patients has not been widely adopted.
The “chloride theory” of fluid retention could be expanded (96-110, 133-144). I am unfamiliar with word “Chlorion” and that should be explained.
Reply We have modified this section.
Ultrafiltration (118-131) remains a niche treatment. Are there advantages and data supporting its use?
Reply We have modified this section.
The paragraph 159-167 should be re-written.
Reply We have modified this section.
The report is well referenced.
Reviewer 2 Report
The review paper by Christina Chrysohoou addressed fluid and salt balance and the role of nutrition in heart failure. Although timely and well-written, there is a need to go through the manuscript and improve the language, text, and grammar. Some typographic and orthographic errors should be corrected throughout the whole manuscript.
Author Response
The review paper by Christina Chrysohoou addressed fluid and salt balance and the role of nutrition in heart failure. Although timely and well-written, there is a need to go through the manuscript and improve the language, text, and grammar. Some typographic and orthographic errors should be corrected throughout the whole manuscript.
We would like to Thank the Reviewer for the useful remarks
Round 2
Reviewer 1 Report
This report is much improved. However, it is still difficult to read in certain sections and requires additional editing. In particular, 55-61 should be revised. The information is correct but the wording is confusing.
Similarly, 75-81 could be clearer.
Lines 124-150 should be 2 paragraphs (perhaps separated at line 133).
Word usage is awkward at times but it does not interfere with interpretation.
Author Response
This report is much improved. However, it is still difficult to read in certain sections and requires additional editing.
In particular, 55-61 should be revised. The information is correct but the wording is confusing.
Similarly, 75-81 could be clearer.
Lines 124-150 should be 2 paragraphs (perhaps separated at line 133).
Word usage is awkward at times but it does not interfere with interpretation.
We would like to Thank the Reviewer for the attention given to this article that helped us to improve it.
We have corrected all the indicated sentences according to the suggestions .
